# Relationship Between the Occurrence of Depression and *DROSHA* (rs6877842, rs10719) and *XPO5* (rs11077) Single-Nucleotide Polymorphisms in the Polish Population: A Case–Control Study

**DOI:** 10.3390/ijms252212204

**Published:** 2024-11-14

**Authors:** Mateusz Kowalczyk, Edward Kowalczyk, Monika Talarowska, Ireneusz Majsterek, Maciej Skrzypek, Tomasz Popławski, Monika Sienkiewicz, Anna Wiktorowska-Owczarek, Paulina Sokołowska, Marta Jóźwiak-Bębenista

**Affiliations:** 1Babinski Memorial Hospital, ul. Aleksandrowska 159, 91-229 Lodz, Poland; mateuszjerzykowalczyk@gmail.com; 2Department of Pharmacology and Toxicology, Medical University of Lodz, ul. Zeligowskiego 7/9, 90-752 Lodz, Poland; edward.kowalczyk@umed.lodz.pl (E.K.); anna.wiktorowska-owczarek@umed.lodz.pl (A.W.-O.); paulina.sokolowska@umed.lodz.pl (P.S.); 3Department of Clinical Psychology and Psychopathology, Institute of Psychology, University of Lodz, ul. Scheiblerow 2, 90-128 Lodz, Poland; monika.talarowska@now.uni.lodz.pl; 4Department of Clinical Chemistry and Biochemistry, Medical University of Lodz, ul. Mazowiecka 5, 92-215 Lodz, Poland; ireneusz.majsterek@umed.lodz.pl (I.M.); maciej.skrzypek@umed.lodz.pl (M.S.); 5Department of Microbiology and Pharmaceutical Biochemistry, Medical University of Lodz, ul. Mazowiecka 5, 92-215 Lodz, Poland; tomasz.poplawski@umed.lodz.pl; 6Department of Pharmaceutical Microbiology and Microbiological Diagnostics, Medical University of Lodz, ul. Muszynskiego 1, 90-151 Lodz, Poland; monika.sienkiewicz@umed.lodz.pl

**Keywords:** major depressive disorder (MDD), single-nucleotide polymorphism, *DROSHA*, *XPO5*, miRNA machinery genes

## Abstract

Although the epidemiology and symptoms of major depressive disorder (MDD) have been well-documented, the etiology and pathophysiology of the disease have not yet been fully explained. Depression arises from intricate interplay among social, psychological, and biological factors. Recently, there has been growing focus on the involvement of miRNAs in depression, with suggestions that abnormal miRNA processing locally at the synapse contributes to MDD. Changes in miRNAs may result from altered expression and/or function of the miRNA biogenesis machinery at the synapse. The aim of our research was to assess the relationship between the occurrence of depression and single-nucleotide polymorphisms (SNP) in the following genes in the Polish population: *DROSHA* (rs6877842; rs10719) and *XPO5* (rs11077). This study involved 200 individuals, including 100 with depressive disorders in the study group (SG) and 100 healthy people without MDD in the control group (CG). All participants were unrelated native Caucasian Poles from central Poland. Blood samples were collected to evaluate the single-nucleotide polymorphism of the genes. Findings indicated that within our patient cohort, the risk of depression is increased by polymorphic variants of the rs10719/*DROSHA* and rs11077/*XPO5* genes and lowered by rs6877842/*DROSHA*. Our study sheds light on the understanding of the genetic basis of depression, which can be used in the rapid diagnosis of this disease.

## 1. Introduction

The epidemiology and symptomatology of depression are well-documented in the existing literature. However, the etiology and pathophysiology of major depressive disorder (MDD) remain incompletely understood. It is established that depression results from a complex interplay of social, psychological, and biological factors. Among the most significant risk factors identified to date is exposure to environmental stressors, particularly traumatic events during early life, which notably contribute to MDD. Adverse environmental stimuli can induce stable changes in gene expression in otherwise healthy individuals, thereby promoting the onset of depression via epigenetic mechanisms. Several primary epigenetic mechanisms have been extensively studied in relation to MDD, including DNA methylation, histone modification, chromatin remodeling, and RNA modulation by non-coding RNAs, such as microRNA (miRNA) and long non-coding RNA (lncRNA) [1,2]. Recently, miRNAs have gained substantial attention as physiological markers of disease progression in patients with clinical depression. In 2018, Yuan and colleagues conducted a meta-analysis examining the expression levels of miRNAs in peripheral tissues of individuals with MDD, comparing these findings to those in control subjects [3]. Their analysis revealed notable alterations in 178 distinct miRNAs relative to the control group. Among these, however, only miRNA-132 demonstrated consistently reproducible evidence, which was confirmed across four independent studies. According to Yoshino et al. (2021), the aberration in individuals with MDD may lie not in the specific type of miRNA but rather in the process by which it is produced [4].

The biogenesis of microRNAs involves two distinct stages. The first phase takes place in the cell nucleus, while the second occurs in the cytoplasm. The synthesis of primary miRNA (pri-miRNA) is initiated with the aid of RNA polymerase II. This pri-miRNA molecule is about 70 nucleotides in length and has a characteristic “hairpin” structure [5]. The subsequent step in the nuclear maturation phase involves a microprocessor complex, which includes the ribonuclease enzyme Drosha (from the ribonuclease III-Rnase III family) and DGCR8, a double-stranded RNA-binding protein also known as the DiGeorge syndrome critical region 8 protein. Through this processing, the pri-miRNA is converted into precursor miRNA (pre-miRNA). This pre-miRNA is then transported to the cytoplasm by the exportin 5 protein (XPO5), thus initiating the second phase of miRNA maturation [6]. In the cytoplasm, Dicer, along with the pre-miRNA enzyme, binds to the TAR RNA-binding protein (TRBP). This binding results in the production of miRNAs approximately 22 nucleotides in length. The resulting miRNA duplex is associated with the RNA-induced silencing complex (RISC), a multi-protein complex responsible for regulating target gene expression through translational inhibition, mRNA degradation, or deadenylation [7]. Studies by Pong and Gullerov highlight the critical role of the elements involved in miRNA synthesis [8]. Using the HCT116 colon cancer cell line, researchers found that both Drosha and Dicer are crucial for miRNA formation. Following the disruption of Drosha and Dicer, 96.5% and 96% of miRNA species were reduced to less than 10% of their original levels. In contrast, after the removal of XPO5, only 29% of miRNA levels showed reduction.

Given the extensive range of biological processes involved in depression, treatments that focus on a single component of depressive pathophysiology are unlikely to yield effective outcomes. Conversely, therapies that address multiple depression-related disorders may offer significant advantages. Such “master regulators” can support overall homeostasis in neuronal signaling disorders. Consequently, treatments aimed at multiple pathways of depression may alleviate symptoms that are unresponsive to conventional antidepressants. Gene-encoding proteins that participate in miRNA maturation might serve as potential targets in this context. In our research, we sought to determine whether specific single-nucleotide polymorphisms (SNPs) in genes associated with nuclear microRNA processing, namely, *DROSHA* (rs6877842; rs10719) and *XPO5* (rs11077), are linked with depression risk in the Polish population. Our study represents the first examination of this kind in the Polish population and one of the few globally that explores these associations in patients with depression.

## 2. Results

### 2.1. Characteristics of the Analyzed Groups

The mean age of the study participants from both groups (N = 200) was 37 years, the standard deviation (SD) was 11, the minimum age was 19 years, and the maximum age was 81 years. Women outnumbered men in both patients with recurrent depressive disorders (80% in the study group) and healthy subjects (over 70% in the control group). There were 100 participants in the study group, including 80 women and 20 men, whereas the control group consisted of 74 women and 26 men. In total, 154 women (77%) and 46 men (23%) participated in this study.

In the study group, according to the Hamilton Depression Rating Scale (HDRS), about 3% of patients demonstrated very severe symptoms of depressive disorders. In addition, severe symptoms were observed in more than 63% of patients, moderate symptoms were observed in about 21%, and about 12% of patients demonstrated mild symptoms. The study qualifying for the experiment was carried out on the day of inclusion and was assessed according to HDRS as follows: <7—no symptoms of depressive disorders, 8–12—mild symptoms, 13–17—moderate symptoms, 18–29—strong intensification of depressive disorder symptoms, >30—highly intensified symptoms of depressive disorders (Table 1).

### 2.2. Analysis of the Compliance of the Frequency of the Studied Polymorphisms with the Hardy-Weinberg Law

Results of the statistical analysis of deviations from the Hardy–Weinberg equilibrium law are presented in Table 2. The frequency distributions of the analyzed polymorphisms showed a deviation from the Hardy–Weinberg equilibrium law (*p* < 0.05).

### 2.3. Analysis of the Relationship Between Occurrence of Depression and the Studied Polymorphic Variants of miRNA Maturation Genes

In the next part of this study, we present the results of a statistical analysis that assessed the relationship between individual polymorphisms in miRNA maturation genes and the occurrence of depression. Association studies were used to conduct this assessment. These population studies aim to determine whether a specific allele of a particular gene is more common in a group of unrelated affected individuals than in healthy individuals. For each polymorphism, the frequency of individual genotypes is presented with the presence or absence of depression. This type of study can consider four different genetic models: codominant, dominant, recessive, and over-dominant. We used the codominant model as it is the most general model and assumes that each genotype generates a distinct, independent risk. This model compares the heterozygous and homozygous genotypes for the variant allele with homozygous genotypes for the most common allele. We set the reference genotypes and alleles based on data from the SNP PubMed database for the Caucasian European population. We found an association between all studied polymorphisms and depression occurrence (see Table 3). The T/G genotype of the rs11077/XPO5 polymorphism and the A/G genotype of the rs10719/DROSHA polymorphism increased the risk of depression, whereas the G/C and C/C genotypes of the rs6877842/DROSHA polymorphism lowered the risk of depression. Haplotype association with depression was also analyzed, and we found two allele combinations (CTA and CGA) that lowered the risk of depression occurrence. However, the statistical significance of the second combination was reduced after multiple corrections (see Table 4).

Table 3 shows the correlation between depression and the frequency of genotypes of the rs10719/DROSHA, rs6877842/DROSHA, and rs11077/XPO5 polymorphism genotypes. A correlation with depression was observed in the rs10719/DROSHA, rs6877842/DROSHA, and rs11077/XPO5 polymorphisms.

## 3. Discussion

Genetic polymorphism refers to the presence of two or more alleles at a specific locus within a population, each with a frequency exceeding 1% [9,10]. This polymorphism can arise from a single-nucleotide variation (SNP) or involve longer sequence differences, such as insertions, deletions, microsatellite sequence repeats, or even the complete absence of a gene [11]. Among these, SNPs are the most commonly examined. Although SNPs are widespread and linked to various diseases, their exact effects on gene expression, protein interactions, and mechanisms of disease remain incompletely understood. SNPs can occur in both coding and non-coding regions, including introns and regulatory sequences. In coding regions, polymorphisms can lead to a change in amino acid sequence, potentially altering the properties of the resulting protein compared to the wild-type allele, such as reduced enzymatic activity, altered regulator affinity, or different ligand-binding characteristics. Due to these direct effects on protein structure, SNPs in coding regions are often readily associated with diseases. However, the majority of SNPs (around 93%) are located in non-coding regions. Changes in these non-coding sequences can affect transcription factor binding sites, alter gene splicing, influence mRNA maturation, or impact transcript stability. Such modifications can reduce the amount of translated protein, ultimately affecting the phenotypic profile.

Alterations in non-coding sequences, which impact critical interactions between RNA and other biomolecules, may explain the influence of SNPs on phenotypes within 5′ and 3′ UTRs or in non-coding RNAs. RNA inherently interacts with RNA-binding proteins (RBPs), RNA-protein complexes, such as the ribosome and spliceosome, and other RNA molecules [12]. These interactions regulate every stage of the RNA lifecycle, including RNA stability, its localization within the cell, the recruitment of ribosomes to mRNA, and ultimately, the level of protein produced per mRNA transcript. It is, therefore, not surprising that disruptions in these interactions can lead to disease.

Recent research suggests that genes involved in microRNA processing play a significant role in various human disorders. These molecules have potential applications as biomarkers or therapeutic targets for diseases marked by diverse pathological changes. Mutations in genes essential for miRNA biogenesis have been identified in motor neuron diseases, such as spinal muscular atrophy (SMA) and amyotrophic lateral sclerosis (ALS). In SMA motor neurons, *DROSHA* expression levels were found to be decreased, while DGCR8 levels were elevated [13]. Similarly, Merritt et al. reported reduced mRNA and protein expression of RNAse III enzymes *DICER1* and *DROSHA* in 60% and 51% of 111 invasive epithelial ovarian cancer samples, respectively [14]. Low *DICER1* expression was significantly correlated with advanced tumor stages (*p* = 0.007), whereas low *DROSHA* expression was associated with suboptimal surgical cytoreduction outcomes (*p* = 0.02). Tumor samples exhibiting both high *DICER1* and *DROSHA* expression were linked to improved median survival rates.

DROSHA was identified as a key factor in pri-miRNA processing over ten years ago. Beyond its role in nuclear pri-miRNA processing, DROSHA also participates in ribosomal RNA biogenesis, where its inhibition causes an accumulation of 5.8S rRNA precursors [15]. Despite this accumulation, rRNA maturation levels show minimal changes following DROSHA depletion, suggesting that DROSHA’s role in rRNA maturation may be relatively minor. One of DROSHA’s best-known functions outside the miRNA pathway is the post-transcriptional destabilization of mRNA [16]. DROSHA-mediated cleavage aids in clearing certain mRNA subsets in progenitor cells, which is crucial for determining cell fate and differentiation [17]. In embryonic neural stem cells, DROSHA destabilizes the mRNA of the proneural factor Neurogenin2 (Ngn2) by cleaving pri-miRNA-like hairpins within its 3′-UTR. The loss of this regulation largely accounts for the premature differentiation seen in progenitors lacking DROSHA, as the overexpression of Ngn2 produces a similar effect, while the dual depletion of DROSHA and Ngn2 restores the normal progenitor state. Another DROSHA target, Tbr1 mRNA, is essential in neuronal progenitors, where its downregulation is vital for corticogenesis [17]. In adult hippocampal stem cells, DROSHA also targets NF1B mRNA to inhibit oligodendrogenesis while promoting neurogenesis.

A similar process occurs in hematopoietic stem cells, where the DROSHA-driven destabilization of *Myl9* and *Todr1* mRNAs is essential for the development of dendritic cells and for myelopoiesis. Beyond these significant biological examples, numerous other mRNAs also undergo DROSHA-mediated cleavage [18,19]. This cleavage leads to their destabilization; however, when DROSHA is depleted, the degree of de-repression of these mRNAs is relatively modest compared to that of pri-miRNAs. In vitro, mRNA hairpins are cleaved by DROSHA with less efficiency than pri-miRNAs [19].

Polymorphisms in the *DROSHA* gene may influence the efficiency of pri-miRNA processing into pre-miRNA, which is a critical step in miRNA biogenesis. DROSHA, which is a key ribonuclease in this process, is essential for the maturation of miRNAs, which are involved in various processes relevant to neuronal plasticity, mood regulation, and stress response [20]. Disruptions in miRNA processing could lead to the altered expression of miRNAs associated with neural pathways, potentially contributing to the pathophysiology of depression.

For instance, miR-124, which is known to play a pivotal role in neurogenesis, shows reduced expression in individuals with depression [21]. Additionally, miR-16, which regulates serotonin pathways, is also dysregulated in depressive disorders [22]. Polymorphisms in *DROSHA*, such as rs10719 and rs6877842, may impact the expression of these miRNAs by altering miRNA maturation, which, in turn, could disrupt the regulation of key genes involved in stress response and neuronal signaling.

Moreover, SNPs in miRNA-processing genes have been shown to impact the functionality and stability of specific miRNAs, further influencing susceptibility to psychiatric disorders, including depression [23]. These findings suggest that *DROSHA* polymorphisms could modulate depressive symptoms by interfering with the miRNA-dependent regulation of gene networks that are critical for mood stability and response to environmental stressors (Figure 1).

In our study, we identified a connection between rs11077 in *XPO5* and an increased risk of depression. Exportin 5 (XPO5) is the transport protein responsible for moving precursor miRNAs to the cytoplasm [6]. XPO5 is widely recognized as a mediator of nuclear export for both siRNA and miRNA. Through its Ran-guanosine triphosphate (GTP)-dependent double-stranded RNA-binding properties, XPO5 facilitates the transport of pre-miRNA from the nucleus to the cytoplasm in a GTP-dependent manner. Once exported, pre-miRNAs continue their maturation process, ultimately becoming functional miRNAs within cells [6]. The removal of XPO5 may lead to an overall reduction in miRNA expression [24]. Findings by Wen, J. et al. [25] indicate that the *XPO5* miRNA-SNP rs11077 could serve as a potential biomarker for predicting thyroid neoplasms. Conversely, other researchers report that SNP rs11077 in *XPO5*, located within the 3′ untranslated region (3′ UTR) of the miRNA-processing gene, enhances the chemotherapy response in metastatic colon cancer. The disruption of XPO5 by siRNAs has been shown to decrease cell proliferation, hinder invasion, induce G1-S cell cycle arrest, and downregulate key oncogenic miRNAs [26]. Additionally, the rs11077 genetic variant in *XPO5* is linked to an increased risk of noise-induced hearing loss in the Chinese population [27].

The relationship we observed between rs 11077/*XPO5* and depression may be due to the fact that miRNA-related single-nucleotide polymorphisms (miR-SNPs), which are defined as single-nucleotide polymorphisms (SNPs) in miRNA genes, the miRNA binding site, and the miRNA-processing mechanism can modulate miRNA and target gene expression. As a result, miR-SNPs can contribute to the risk of many diseases. It turns out that polymorphism of the rs11077/*XPO5* gene also induces depression, but it is still unknown how this is done.

One potential mechanism may involve miR-16, which is a microRNA associated with serotonin signaling that plays a key role in mood regulation. miR-16 regulates the expression of the serotonin transporter, and its dysregulation has been implicated in depressive disorders [22]. Given that XPO5 is crucial for the nuclear export of pre-miRNAs, including miR-16, polymorphisms such as rs11077 could disrupt this export process, potentially leading to altered serotonin signaling and contributing to the pathogenesis of depression (Figure 1). To further substantiate this link, future studies could assess serotonin transporter expression levels in peripheral blood platelets. As the serotonin transporter is expressed both centrally and peripherally, such measurements could offer valuable insights and allow for a direct comparison of the central and peripheral effects of *XPO5* polymorphisms. Highlighting the role of miR-16 in this context could enhance our understanding of the molecular mechanisms underlying depression risk.

### Haplotype Association with Depression

Genetic polymorphisms rarely play a direct role in causing a disease. Instead, they tend to influence an individual’s susceptibility or resistance to the factors that trigger the disease or affect its progression. This is because the phenotype is shaped not only by the genotype but also by its interaction with the environment. It is important to note that gene–environment interactions are highly complex, as the body is constantly exposed to a variety of both beneficial and harmful factors. Our health is impacted by agents that can either harm or challenge the system, as well as by those that support repair and detoxification processes. Additionally, our genotype comprises a vast mosaic of polymorphic genes, each of which can affect susceptibility to disease in modifiable ways. Furthermore, many genes exhibit pleiotropy, meaning they influence multiple phenotypic traits rather than a single trait, contrary to the traditional definition of a gene’s role. The potential positive or negative impact of a specific polymorphism should often be considered in the context of the body’s current demands and its physiological state. A deeper understanding of these interactions may open pathways for new therapeutic strategies to slow or halt disease progression.

In terms of methodological quality, the greatest limitation of our study was the relatively small population size, as sample size may affect the accuracy of results. To assess the power of the sample size, QUANTO 1.24 (https://keck.usc.edu/biostatistics/software/, accessed on 1 October 2024) was used. We estimated the incidence of depression in Poland at 7.5%, and allele frequencies were obtained from dbSNP (https://www.ncbi.nlm.nih.gov/snp/rs11077; accessed on 1 October 2024). Under these sample power conditions (α = 0.05, twofold risk, and 100 cases), we achieved power values of 0.986 for rs10719, 0.9445 for rs6877842, and 0.895 for rs11077. Therefore, our case–control study was sufficiently powered to detect a correlation, so there is no immediate need to increase the sample size. However, a larger study population could further validate and broaden the scope of our findings.

The genotype distribution for analyzed polymorphisms, specifically rs6877842 and rs11077, did not align with Hardy–Weinberg equilibrium (HWE) expectations. HWE is a foundational concept in population genetics, positing that in the absence of external factors, genotype frequencies remain stable across generations. Several factors can disrupt HWE, with genotyping errors often cited as a primary reason, particularly in case–control studies. This assumption stems from the idea that genotype frequencies in a large, randomly mating population should conform to HWE. However, deviations from HWE cannot automatically justify discarding test results based on potential genotyping errors [28]. Genotyping errors are generally infrequent and do not commonly lead to HWE deviation. Moreover, heterozygote frequency is a key indicator; a reduced heterozygosity frequency (LoH) points to factors beyond genotyping errors, such as purifying selection, copy number variation, inbreeding, or population substructure [28,29,30]. Additional verification of genotyping accuracy is recommended in cases in which an excess of heterozygotes (GoH) is observed. In our study, heterozygosity frequency analysis did not show GoH, indicating the absence of genotyping errors. Consequently, the conclusions of our work are scientifically based and facilitate an understanding of the genetic basis of depression.

## 4. Materials and Methods

This study spanned from January 2019 to December 2020 and involved the examination of 200 individuals—100 diagnosed with depressive disorders in the study group (SG) and 100 healthy people without MDD in the control group (CG). All participants were native, unrelated Caucasian Poles residing in central Poland. Blood samples were collected from both the patients and subjects in the control group to assess polymorphisms in the following genes: *DROSHA* (rs6877842, rs10719) and XPO5 (rs11077). Participants with chronic inflammatory diseases or neurological or organic disorders, who were undergoing oncological treatment, who were metabolically unbalanced, who were suffering from injuries (including head injuries), who had autoimmune diseases, who had an addiction to psychotropic or other drugs, and who refused to give consent for participation in this study were excluded from it. Each participant from both the study and control groups also completed a Hamilton Depression Scale form, facilitating the exclusion of individuals with abnormal results from the control group.

Each of the respondents expressed his or her consent in writing to participate in this study in accordance with the protocol approved by the Bioethics Committee of the Medical University of Lodz, No.: RNN/402/18.

### 4.1. Study Group

The study group consisted of 100 people hospitalized due to recurrent depressive disorders at the Department of Adult Psychiatry of the Medical University of Lodz and at the Specialist Psychiatric Healthcare Complex in Lodz of the J. Babinski Hospital.

Patients were qualified for the study group on the basis of diagnostic criteria included in the International Statistical Classification of Diseases and Related Health Problems (ICD-10) for recurrent depressive disorders (F33.0–F33.8). All patients underwent examination upon admission to the hospital.

### 4.2. Control Group

The control group (GP) consisted of 100 healthy people with a negative family history of mental illness and a score of 0–7 points on the Hamilton Depression Scale. The exclusion criteria were the same as for the study group.

### 4.3. DNA Isolation

Blood samples were collected in anticoagulant EDTA tubes (Sarstedt, Nümbrecht, Germany) for DNA isolation. Subsequently, DNA isolation was carried out using the QI-Aamp DNA Blood Mini Kit (Qiagen, Chatsworth, CA, USA) following the manufacturer’s protocol provided with the kit. Following the isolation procedure, DNA samples were preserved at −20 °C in TE buffer (pH 8.0). The concentration and purity of the DNA preparations were quantified by measuring absorbance at 260 and 280 nm on a Nanodrop Spectrophotometer (Thermo Scientific, Waltham, MA, USA).

### 4.4. Determination of Single-Nucleotide Polymorphisms

The frequency of polymorphic variants of genes *DROSHA* (rs6877842; rs10719) and *XPO5* (rs11077) was determined using the TaqMan^®^ SNP Genotyping Assays and TaqMan Universal PCR Master Mix, No AmpErase UNG (Applied Biosystems, Foster City, CA, USA). These kits contained primers and fluorescently labeled molecular probes that enabled us to read the genotype during a real-time DNA polymerase chain reaction analysis. The analyses were conducted in accordance with the manufacturer’s recommendations. The reaction was performed using the Stratagene Mx3005p system (Agilent Technologies, Santa Clara, CA, USA).

### 4.5. Statistical Analysis

Statistical analysis was performed using the STATISTICA software (TIBCO Software Inc. (2017). Statistica (data analysis software system), version 13.3 http://statistica.com, accessed on 1 October 2024). The Chi-squared (χ2) test was used to assess the consistency of the genotype distribution with the Hardy–Weinberg (HWE) distribution. The significance of differences between the frequencies of genotypes in both the patients and controls was assessed. As we conducted multiple tests (three SNPs), we adjusted the threshold for statistical significance to 0.005. The statistical analysis also assessed the risk of an event (odds ratio—OR) and presented the confidence interval (95% CI), which was calculated with the use of the logistic regression model.

## 5. Conclusions

We believe that SNPs in *DROSHA* (rs10719 and rs6877842) and *XPO5* (rs11077) are associated with an increased risk of depression. In light of recent findings from a meta-GWAS on MDD, which offers a comprehensive view of genetic variants associated with depression across diverse populations, our study adds unique insights specific to the Polish population [31]. Nevertheless, it is advisable to confirm our observations in a larger cohort of patients to strengthen these findings. Future research could focus on expanding the study population to include patients experiencing their first depressive episode, thus allowing for a comparative analysis with those affected by recurrent depression. Such an approach may provide valuable insights into whether these specific SNPs in *DROSHA* and *XPO5* contribute to recurrent episodes of depression, thereby offering insights into the genetic factors that influence the likelihood of relapse. Understanding this relationship may enhance our ability to address the mechanisms underlying depression’s persistence and recurrence.

## Figures and Tables

**Figure 1 ijms-25-12204-f001:**
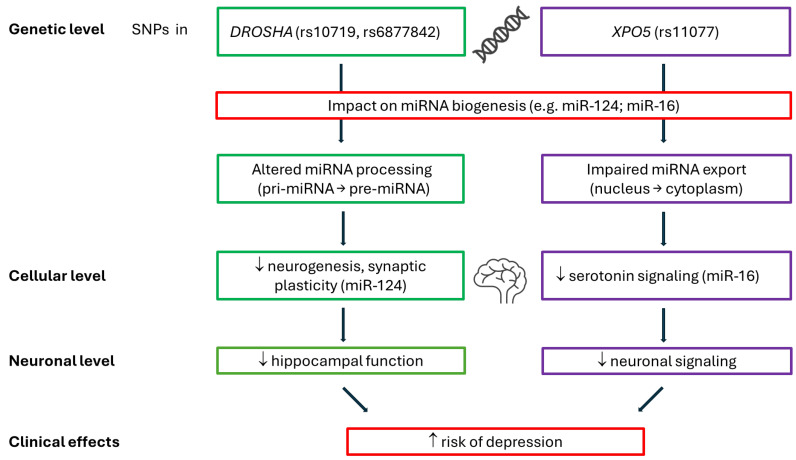
Link between SNPs in *DROSHA* and *XPO5* genes, their impact on miRNA biogenesis, neurogenesis, and synaptic plasticity and the associated risk of depression. ↑ increase; ↓ decrease.

**Table 1 ijms-25-12204-t001:** Severity of depression in the study group according to HDRS score criteria.

Group	Gender	Participants	Very Severe Symptoms (>30 HDRS)	Severe Symptoms (18–29 HDRS)	Moderate Symptoms (13–17 HDRS)	Mild Symptoms (8–12 HDRS)	No Symptoms (<7 HDRS)
Study Group	Women	80	3%	63%	21%	12%	-
	Men	20					
Control Group	Women	74	-	-	-	-	-
	Men	26					

**Table 2 ijms-25-12204-t002:** Analysis of the consistency of the genotype distribution in the analyzed miRNA gene maturation polymorphisms.

Polymorphism/Gene	*p* *		
	Totality	Control Group	Study Group
rs6877842/*DROSHA*	<0.0001	<0.0001	<0.0001
rs10719/*DROSHA*	0.009	0.0001	1
rs11077/XPO5	0.02	0.0002	0.84

* *p* < 0.05—in compliance with HWE.

**Table 3 ijms-25-12204-t003:** Comparison of depression with the frequency of rs10719/DROSHA, rs6877842/DROSHA, and rs11077/XPO5 polymorphism genotypes. Bold text highlights genotypes associated with an increased risk of depression (red) or a decreased risk of depression (green).

Polymorphism/Gene	Genotype	Control Group	Study Group	OR (95% CI)	*p*-Value
rs10719/*DROSHA*	A/A	67 (67%)	43 (43%)	1.00	---
	** A/G **	**21** (**21%**)	**45** (**45%**)	** 3.34 ** (**1.75–6.35**)	**0.0002**
	G/G	12 (12%)	12 (12%)	1.56 (0.64–3.78)	0.33
rs6877842/*DROSHA*	G/G	21 (21%)	48 (48%)	1.00	---
	** G/C **	**25** (**25%**)	**20** (**20%**)	** 0.35 ** (**0.16–0.76**)	**0.008**
	** C/C **	**54** (**54%**)	**32** (**32%**)	** 0.26 ** (**0.13–0.59**)	**0.0001**
rs11077/*XPO5*	T/T	44 (44%)	26 (26%)	1.00	---
	** T/G **	**30** (**30%**)	**52** (**52%**)	** 2.93 ** (**1.51–5.68**)	**0.0014**
	G/G	26 (26%)	22 (22%)	1.43 (0.68–3.02)	0.35

**Table 4 ijms-25-12204-t004:** Haplotype association with depression (n = 200, crude analysis).

	rs6877842	rs.11077	rs10719	Freq	OR (95% CI)	*p*-Value
1	G	T	A	0.2239	1.00	---
2	C	T	A	0.2226	0.30 (0.15–0.58)	0.0005
3	G	G	A	0.166	0.74 (0.35–1.58)	0.44
4	C	G	G	0.1234	1.20 (0.54–2.65)	0.66
5	C	G	A	0.1025	0.19 (0.07–0.56)	0.0029 *
6	C	T	G	0.094	0.75 (0.34–1.69)	0.49
7	G	G	G	0.0531	3.97 (0.81–19.37)	0.09
8	G	T	G	0.0145	0.10 (0.01–1.35)	0.085

* No statistically important after multiple corrections.

## Data Availability

Data is contained within the article.

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
