# Peer review of "Relationship Between the Occurrence of Depression and *DROSHA* (rs6877842, rs10719) and *XPO5* (rs11077) Single-Nucleotide Polymorphisms in the Polish Population: A Case–Control Study"

_ijms, 2024, doi:10.3390/ijms252212204_

Round 1
Reviewer 1 Report
Comments and Suggestions for Authors
Manuscript IJMS "Relationship between the occurrence of depression and 2 DROSHA (rs6877842, rs10719) and XPO5 (rs11077) single nucleotide polymorphisms in the Polish population: a case-control-study" by Mateusz Kowalczyk et al.
In this manuscript, the authors are interested in major depressive disorder (MDD) with a specific focus on the involvement of miRNA maturation in depression. For this purpose, they assess the relationship between depression and single nucleotide polymorphisms (SNP) in two genes involved in miRNA maturation, DROSHA (rs 6877842; rs 10719) and XPO5 (rs11077) in the Polish population. Using a cohort of 200 individuals, comprising 100 with depressive disorders in the study group and 100 without MDD in the control group, they report that within the patient cohort, the risk of depression is increased by polymorphic variants rs10719 in DROSHA gene and rs11077 in XPO5 gene. The authors conclude that this study sheds light on understanding of the genetic basis of depression: changes in miRNAs may result from altered expression and/or function of the miRNA biogenesis machinery.
Although it has been proposed that in individuals with MDD, the abnormality lies not in the type of miRNA, but in the process by which it is formed, e.g. maturation by the Drosha ribonuclease and/or transfer to the cytoplasm by the exportin 5 transport protein (XPO5), this work presents interest for the readers. However, there some points that should be clarified before publication.
1-In their analysis, the authors found deviations from the Hardy-Weinberg equilibrium, which is quite puzzling. Furthermore, the study group consisted of only 100 people hospitalized due to a recurrent depressive disorder. As correctly discussed by the authors, "the greatest limitation of our study was a relatively small size of the population. Sample size may affect the accuracy of results". It therefore questionable whether the results obtained by the authors care correct, unless they could increase sample size.
2-An important missing information is the mechanism linking the analyzed polymorphic variants with depression. Genes encoding proteins involved in miRNA maturation could represent master regulators and thus be involved in general neuronal signaling and its defects such as depression. DROSHA-mediated cleavage contributes to clearance of a subset of mRNAs in progenitor cells, which appears to be essential for cell fate determination and differentiation. In adult hippocampal stem cells, DROSHA has been shown to promote neurogenesis. A similar mechanism may operate in hematopoietic stem cells. Furthermore, XPO5 alterations may result in reduced cell proliferation, impaired invasion, and cell cycle arrest. It would therefore be interesting to test in the identified patients bearing these SNPS to study if there are defects in stem cells and/or to evaluate their hippocampus functions.
Many typos e.g. line77: RISC com-plex
Author Response
Dear Reviewer,
We would like to sincerely thank you for your valuable feedback and insightful comments regarding our manuscript. We have carefully considered each of your suggestions and made the necessary revisions. Please find our responses to your comments below:
1-In their analysis, the authors found deviations from the Hardy-Weinberg equilibrium, which is quite puzzling. Furthermore, the study group consisted of only 100 people hospitalized due to a recurrent depressive disorder. As correctly discussed by the authors, "the greatest limitation of our study was a relatively small size of the population. Sample size may affect the accuracy of results". It therefore questionable whether the results obtained by the authors care correct, unless they could increase sample size.
Hardy-Weinberg Equilibrium (HWE) Deviation: In our study, we aimed to explain the observed deviation from Hardy-Weinberg equilibrium (HWE) by referencing findings from other researchers. As noted by Royo (2020), “deviation from HWE should not be used as definitive grounds to reject test results due to purported genotyping errors.” First, genotyping errors are generally rare and do not typically lead to HWE deviations. Furthermore, the number of observed heterozygotes is an important parameter that may suggest the involvement of genotyping errors in HWE deviations.
The reduced heterozygosity (LoH) observed in our study may result from factors other than genotyping errors, such as purifying selection, copy number variation, inbreeding, or population substructure (Obirikorang et al., 2024; Royo, 2020; Santos et al., 2019). In human studies, none of these factors can be excluded. The authors recommend conducting additional analysis to verify genotyping accuracy if an excess of heterozygotes (GoH) is observed. Our analysis of heterozygosity frequencies did not indicate any GoH, thereby suggesting an absence of genotyping errors.
It is also worth noting, as highlighted by Santos et al. (2019), that deviation from HWE may serve as supportive evidence for the association of a specific genotype with a disease. In our case, this could suggest a significant association between the studied polymorphisms and depression.
Sample Size: We conducted the power of our study analysis, at the request of another reviewer. The power of the sample size was estimated using a QUANTO 1.24 (https://keck.usc.edu/biostatistics/software/). We estimated incidence of depression in Poland at 7.5%. Allele frequency were obtained from dbSNP (https://www.ncbi.nlm.nih.gov/snp/rs11077; October 2024). When the conditions of the sample power (α = 0.05, risk = two-fold, the number of cases = 100) were employed, we had 0.986 for rs10719, 0.9445 for rs6877842 and 0.895 for rs11077. Therefore, our case–control study was sufficiently powerful to study a positive correlation so there is no need to increase the group size. We included this in the manuscript.
2-An important missing information is the mechanism linking the analyzed polymorphic variants with depression. Genes encoding proteins involved in miRNA maturation could represent master regulators and thus be involved in general neuronal signaling and its defects such as depression. DROSHA-mediated cleavage contributes to clearance of a subset of mRNAs in progenitor cells, which appears to be essential for cell fate determination and differentiation. In adult hippocampal stem cells, DROSHA has been shown to promote neurogenesis. A similar mechanism may operate in hematopoietic stem cells. Furthermore, XPO5 alterations may result in reduced cell proliferation, impaired invasion, and cell cycle arrest. It would therefore be interesting to test in the identified patients bearing these SNPS to study if there are defects in stem cells and/or to evaluate their hippocampus functions.
The information about the mechanism linking the analyzed polymorphic variants with depression, as well as further research plans, has been added in the manuscript in red.
We agree that the proposed studies would certainly enrich our knowledge regarding the molecular factors of depression and the potential role of XPO5. Nevertheless the Bioethics Committee of the Medical University of Lodz approved our study under consent No. RNN/402/18, which focused exclusively on selected gene polymorphisms involved in microRNA processing in depressed patients. Participants consented solely for this research.
Given the promising results, we are eager to expand our inquiry. We have approached the Bioethics Committee for new consent and plan to collaborate with researchers from other centers across Poland, beyond Lodz. By doing so, we aim to gather a larger population for a comprehensive analysis of the relationship between gene polymorphisms and demographic data. Importantly, we want to include patients experiencing their first depressive episode, allowing us to juxtapose their outcomes with those of individuals facing recurrent depression. Furthermore, we plan to seek to explore gene polymorphisms associated with xenobiotic metabolism, potentially addressing the reasons for antidepressant treatment ineffectiveness. Biochemical tests in these patients will also enhance our findings.
To realize this ambitious plan, we need to secure additional funding for our research. Publishing the studies currently revised will be crucial; it will showcase the validity of our research idea and bolster our applications for new funding opportunities.
Reviewer 2 Report
Comments and Suggestions for Authors
In the present manuscript, authors aim to assess the relationship between occurrence of depression and single nucleotide polymorphisms (SNP) in the following genes: DROSHA (rs 6877842; rs 31 10719) and XPO5 (rs11077) in the Polish population, in 200 people: 100 cases and 100 controls. It is an interesting topic. However, as the authors have commented, this is an original work in which they report that the presence of a polymorphism of two genes: DROSHA (rs6877842, rs10719) and XPO5 (rs11077) could be associated with depression. Although there are different reports on the Association with other pathologies such as endometriosis, breast cancer risk, and thyroid cancer and microRNA-Related Gene XPO5 449 Rs11077 Polymorphism.
¿Could you comment on how or where the idea to look for the association with depression of these DROSHA (rs6877842, rs10719) and XPO5 (rs11077) single nucleotide polymorphisms came from?
¿Could you comment something about the mechanisms linking the analyzed polymorphic variants of DROSHA genes and depression?
Some observations and questions
a) ¿Was there any particular reason why the number of female participants was higher in both groups?
b) Due to the majority number of women in both study groups; ¿the authors could identify whether the women correspond to the 3% of the very severe symptoms from the study group? Or from both?
c) ¿Could the authors consider producing a table showing the HDRS results by study group and gender?
d) In the table 2 could indicate in bold the result the authors want to show about the correlation of depression with the frequency of genotypes of the rs10719/DROSHA, rs6877842/DROSHA and rs11077/XPO5 polymorphism genotypes. I mean, the T/G genotype of the rs11077/XPO5 polymorphism and the A/G genotype of the 142 rs10719/DROSHA polymorphism increase the risk of depression, whereas the G/C and C/C genotypes of the rs6877842/DROSHA polymorphism lowered the risk of depression.
Some minimal errors that should be checked.
1) In page 1, in the abstract, in line 33
and 100 without MDD in the control
As a suggestion, you could add: and 100 healthy people without MDD in the control
2) In page 2, line 68
miRNA is ap-proximately 70 nucleotides
As a suggestion, please removed the dash in the word ap-proximately.
miRNA is approximately 70 nucleotides
3) In page 2, line 77
to the RISC com-plex (RNA-induced silencing complex).
As a suggestion, please removed the dash in the word com-plex.
to the RISC complex (RNA-induced silencing complex).
4) In page 3, line 106
according to on the HDRS (Hamilton Depression Scale) scale,
As a suggestion, please add the word Rating in the acronym (Hamilton Depression Scale)
Hamilton Depression Rating Scale.
5) In page 7, line 259
Results obtained by [24] suggest that XPO5…
As a suggestion, please add the name of the researcher instead of the reference [24]
Results obtained by Wen, J.et al suggest that XPO5…
6) In page 8, line 317
100 without MDD in the control group (CG).
As a suggestion, please add healthy people in the sentence
100 healthy people without MDD in the control.
Author Response
Dear Reviewer,
Thank you very much for your thorough review and valuable feedback on our manuscript. We greatly appreciate your insightful comments and suggestions. Below, we provide our responses to each of your points.
- Could you comment on how or where the idea to look for the association with depression of these DROSHA (rs6877842, rs10719) and XPO5 (rs11077) single nucleotide polymorphisms came from?
The idea to investigate the relationship between depression and these particular polymorphisms stems from growing evidence that genes involved in miRNA processing play a critical role in regulating gene expression in the brain, which is essential for neuronal plasticity, neurogenesis, and stress response pathways. Alterations in miRNA biogenesis have been implicated in the pathophysiology of various psychiatric disorders, including depression. Previous studies have shown associations between these genes and other pathologies, such as endometriosis, breast cancer, and thyroid cancer, as well as miRNA-related processes. Given this background, we hypothesized that DROSHA and XPO5 polymorphisms could influence susceptibility to depression by disrupting miRNA processing, which in turn could impair neuronal function and contribute to the development of depressive symptoms.
- Could you comment something about the mechanisms linking the analyzed polymorphic variants of DROSHA genes and depression?
The DROSHA gene encodes a ribonuclease critical for the initial processing of pri-miRNAs into pre-miRNAs, a key step in miRNA biogenesis. miRNAs regulate the expression of genes involved in synaptic plasticity, neuronal development, and stress responses. Disruptions in miRNA processing due to polymorphisms in DROSHA could lead to altered gene expression, potentially affecting neural circuits responsible for mood regulation and stress response. Studies suggest that impaired neurogenesis and synaptic plasticity, particularly in the hippocampus, are associated with depression. The rs6877842 and rs10719 polymorphisms in DROSHA may alter the expression or function of DROSHA, potentially increasing the risk of developing depression by influencing these molecular pathways (Moszyńska et al., 2017; Gregory et al., 2005).
We have included information about these mechanisms and the proposed link to depression in the manuscript, with modifications highlighted in red and referenced in Figure 1.
Some observations and questions
a) Was there any particular reason why the number of female participants was higher in both groups?
The higher number of female participants reflects the general demographic trend, as depression is more prevalent in women than men. Epidemiological data consistently show that women are twice as likely to develop depression as men. Our study reflects this gender distribution naturally, as it is in line with the observed higher incidence of depression in women.
b) Due to the majority number of women in both study groups; ¿the authors could identify whether the women correspond to the 3% of the very severe symptoms from the study group? Or from both?
In our study, women indeed represented the majority in both the study group with recurrent depressive disorders (80%) and the control group (over 70%). However, we did not conduct a specific analysis to determine whether the women correspond to the subset with very severe symptoms, as the severity of symptoms was evaluated across the whole study population. We acknowledge that a gender-specific analysis of symptom severity could provide additional insights and plan to consider this aspect in future studies.
c) Could the authors consider producing a table showing the HDRS results by study group and gender?
A table has been added in accordance with the suggestion.
d) In the table 2 could indicate in bold the result the authors want to show about the correlation of depression with the frequency of genotypes of the rs10719/DROSHA, rs6877842/DROSHA and rs11077/XPO5 polymorphism genotypes. I mean, the T/G genotype of the rs11077/XPO5 polymorphism and the A/G genotype of the 142 rs10719/DROSHA polymorphism increase the risk of depression, whereas the G/C and C/C genotypes of the rs6877842/DROSHA polymorphism lowered the risk of depression.
Table 2 has been revised.
Some minimal errors that should be checked.
1) In page 1, in the abstract, in line 33
and 100 without MDD in the control
As a suggestion, you could add: and 100 healthy people without MDD in the control
2) In page 2, line 68
miRNA is ap-proximately 70 nucleotides
As a suggestion, please removed the dash in the word ap-proximately.
miRNA is approximately 70 nucleotides
3) In page 2, line 77
to the RISC com-plex (RNA-induced silencing complex).
As a suggestion, please removed the dash in the word com-plex.
to the RISC complex (RNA-induced silencing complex).
4) In page 3, line 106
according to on the HDRS (Hamilton Depression Scale) scale,
As a suggestion, please add the word Rating in the acronym (Hamilton Depression Scale)
Hamilton Depression Rating Scale.
5) In page 7, line 259
Results obtained by [24] suggest that XPO5…
As a suggestion, please add the name of the researcher instead of the reference [24]
Results obtained by Wen, J.et al suggest that XPO5…
6) In page 8, line 317
100 without MDD in the control group (CG).
As a suggestion, please add healthy people in the sentence
100 healthy people without MDD in the control.
Thank you very much for your careful review. All minimal errors have been revised according to your suggestions.
Reviewer 3 Report
Comments and Suggestions for Authors
My comments to the reviewers are as follows:
1. The HWE deviation is quite significant. These SNPs (at least the one located in DROSHA) are likely to be in high LD with each other. If so, what is the LD between them? If they're in a high LD, why were all SNPs assessed independently, as this would be redundant?
2. The description of the employed statistical analyses needs to be expanded. While not necessarily unusual, all 3 SNPs have relatively large effect sizes (as measured by the ORs). This is unusual for psychiatric phenotypes, including MDD. Have the authors investigated any additional demographic (since the sample description is scant) data that could have confounded the results?
3. The interest and, more importantly, the confidence in the authors' results would have been substantially increased if they had also tested the impact of the associated SNPs on the expression of DROSHA and XPO5 in their sample.
4. Somewhat a minor comment: I would suggest they remove or at least shorten the text between lines 162 and 217.
Author Response
Dear Reviewer,
We sincerely appreciate the time and effort you have taken to review our manuscript. Your valuable feedback and thoughtful comments are greatly appreciated, and we are grateful for your insightful suggestions. Below, we have provided detailed responses to each of your points.
- The HWE deviation is quite significant. These SNPs (at least the one located in DROSHA) are likely to be in high LD with each other. If so, what is the LD between them? If they're in a high LD, why were all SNPs assessed independently, as this would be redundant?
We calculated LD using SNPstat (https://www.snpstats.net/) . The results are presented below as D' statistic. There is no LD between them.
|
|
rs6877842 |
rs11077 |
rs10719 |
|
rs6877842 |
- |
0.0619 |
0.4757 |
|
rs11077 |
- |
- |
0.2532 |
|
rs10719 |
- |
- |
- |
- The description of the employed statistical analyses needs to be expanded. While not necessarily unusual, all 3 SNPs have relatively large effect sizes (as measured by the ORs). This is unusual for psychiatric phenotypes, including MDD. Have the authors investigated any additional demographic (since the sample description is scant) data that could have confounded the results?
We understood the importance of having a detailed demographic description of our patients, especially given the small size of our study population. Our participants were provided with comprehensive information about genetic testing and the necessity of completing questionnaires on basic demographics and disease progression. We were confident that they would share this vital information without hesitation.
Unfortunately, out of the 100 patients invited to participate, only 67 returned the questionnaire, and merely 48 completed it accurately. This limited response rate severely restricted our ability to analyze any correlations between our results and demographic data. This emphasizes the need for improved engagement in future studies to enhance the robustness of our findings.
- The interest and, more importantly, the confidence in the authors' results would have been substantially increased if they had also tested the impact of the associated SNPs on the expression of DROSHA and XPO5 in their sample.
We agree that the proposed studies would certainly enrich our knowledge regarding the molecular factors of depression and the potential role of DROSHA and XPO5. Nevertheless the Bioethics Committee of the Medical University of Lodz approved our study under consent No. RNN/402/18, which focused exclusively on selected gene polymorphisms involved in microRNA processing in depressed patients. Participants consented solely for this research.
After reviewing our results, we have decided to continue our research and have applied for new ethical approval to include a larger population from different research centers across Poland. This will allow us to conduct more comprehensive analyses, including correlations with demographic data, and to explore additional factors such as xenobiotic metabolism genes, which may be relevant for the efficacy of antidepressant treatment. We also plan to conduct biochemical tests in future studies to further investigate the potential mechanisms underlying these associations.
- Somewhat a minor comment: I would suggest they remove or at least shorten the text between lines 162 and 217.
Thank you for your comment. When writing the manuscript, we considered the potential readership. For geneticists and those familiar with gene polymorphisms, the information in this section might seem basic. However, for readers outside of this field, the section provides valuable context regarding the role of gene polymorphisms. If the journal's editorial team agrees, we are open to shortening or removing the section between lines 162 and 217, as suggested.
Reviewer 4 Report
Comments and Suggestions for Authors
The manuscript presented results of a case-control study that investigated the association between the occurrence of depression and specific single nucleotide polymorphisms (SNPs) in two genes involved in nuclear microRNA processing: DROSHA (specifically rs6877842 and rs10719) and XPO5 (specifically rs11077), within a sample of the Polish population. Although this study can provide important insights in the pathogenesis of depression, I think that the lack of any functional analyses significantly limits the overall relevance and impact of the findings. Here below I report my major concerns that can improve the quality of the manuscript:
- I suggest the authors to perform a relative expression analysis in order to evaluate the mRNA levels of DROSHA and XPO5. This could be useful in determining whether variations in mRNA levels are associated with the occurrence of depression or the investigated SNPs.
- A table summarizing demographic and clinical characteristics of the study population is missing.
- The size of studied population is quite small. I would encourage the authors to perform a power calculation of their study. Indeed, calculating the power of your study will improve the relevance of the results.
- Did you find any differences in the frequency of these investigated SNPs, stratifying the population according to the age, gender and/or kind of depressive disorders?
- How can you speculate the effects of the investigated SNPs? In relation with some specific microRNAs altered in depressive syndromes.
- As the authors stated, patients with depression commonly show disorders of the pituitary, adrenal cortex and other endocrine glands. Therefore, I suggest the authors to correlate their data with levels of certain hormones, such as cortisol, T3, T4, and oxytocin.
Author Response
Dear Reviewer,
We would like to thank you for the careful and thorough reading of this manuscript and for the thoughtful comments and constructive suggestions, which have helped improve its quality. All changes in the revised manuscript are marked in red.
- I suggest the authors to perform a relative expression analysis in order to evaluate the mRNA levels of DROSHA and XPO5. This could be useful in determining whether variations in mRNA levels are associated with the occurrence of depression or the investigated SNPs.
We agree that the proposed studies would certainly enrich our knowledge regarding the molecular factors of depression and the potential role of DROSHA and XPO5. Nevertheless the Bioethics Committee of the Medical University of Lodz approved our study under consent No. RNN/402/18, which focused exclusively on selected gene polymorphisms involved in microRNA processing in depressed patients. Participants consented solely for this research. To effectively compare genotype and expression, we must focus on analyzing expression in brain tissue. Using other tissue types is not just ineffective; it wastes valuable time and resources. Additionally, it's unlikely that the bioethics committee would support the idea of physically burying our patients' brains.
- A table summarizing demographic and clinical characteristics of the study population is missing.
We understood the importance of having a detailed demographic description of our patients, especially given the small size of our study population. Our participants were provided with comprehensive information about genetic testing and the necessity of completing questionnaires on basic demographics and disease progression. We were confident that they would share this vital information without hesitation.
Unfortunately, out of the 100 patients invited to participate, only 67 returned the questionnaire, and merely 48 completed it accurately. This limited response rate severely restricted our ability to analyze any correlations between our results and demographic data. This emphasizes the need for improved engagement in future studies to enhance the robustness of our findings.
- The size of studied population is quite small. I would encourage the authors to perform a power calculation of their study. Indeed, calculating the power of your study will improve the relevance of the results.
We conducted the power of our study analysis. The power of the sample size was estimated using a QUANTO 1.24 (https://keck.usc.edu/biostatistics/software/). We estimated incidence of depression in Poland at 7.5%. Allele frequency were obtained from dbSNP (https://www.ncbi.nlm.nih.gov/snp/; October 2024). When the conditions of the sample power (α = 0.05, risk = two-fold, the number of cases = 100) were employed, we had 0.986 for rs10719, 0.9445 for rs6877842 and 0.895 for rs11077. Therefore, our case–control study was sufficiently powerful to study a positive correlation. We included this in the manuscript.
- Did you find any differences in the frequency of these investigated SNPs, stratifying the population according to the age, gender and/or kind of depressive disorders?
We did not find differences in the frequency of investigated SNPs, stratifying the population according to other covariates.
- How can you speculate the effects of the investigated SNPs? In relation with some specific microRNAs altered in depressive syndromes.
The investigated SNPs in DROSHA and XPO5 may influence the processing and transport of specific miRNAs implicated in mood regulation and stress response, particularly miR-16. The DROSHA gene encodes a ribonuclease essential for the initial processing of pri-miRNAs into pre-miRNAs, a critical step in miRNA biogenesis. Similarly, XPO5 mediates the export of pre-miRNAs from the nucleus to the cytoplasm, where they undergo further processing. Polymorphisms in these genes, such as rs6877842 and rs10719 in DROSHA and rs11077 in XPO5, may alter these functions, affecting the expression and stability of miRNAs, including miR-16, which is linked to serotonin signaling (Gregory et al., 2005; Moszyńska et al., 2017).
miR-16 plays a crucial role in regulating the serotonin transporter, and its dysregulation has been observed in depressive disorders. Given that XPO5 is essential for the export of miR-16, disruptions caused by rs11077 could lead to altered serotonin signaling and potentially contribute to depressive symptoms. Additionally, disturbances in DROSHA-mediated processing could impact other miRNAs involved in neurogenesis and synaptic plasticity, such as miR-124, which is associated with hippocampal function and has reduced expression in depression (Issler & Chen, 2015; Dwivedi, 2014).
These SNPs may therefore interfere with miRNA pathways critical for mood regulation, as represented in our proposed schematic (Figure 1), which outlines a potential mechanism linking these genetic variants to depression risk.
- As the authors stated, patients with depression commonly show disorders of the pituitary, adrenal cortex and other endocrine glands. Therefore, I suggest the authors to correlate their data with levels of certain hormones, such as cortisol, T3, T4, and oxytocin.
We agree that endocrine dysfunctions are common in depression, and analyzing hormonal profiles in conjunction with genetic data could provide valuable insights into the molecular mechanisms underlying depressive disorders.
However, our current study was approved by the Bioethics Committee to focus specifically on genetic polymorphisms, and we did not collect hormonal data from participants. To address this limitation, we are planning future studies to include hormonal assessments in a larger sample. We believe that such analyses could enrich our understanding of the interactions between endocrine and genetic factors in depression.
Round 2
Reviewer 1 Report
Comments and Suggestions for Authors
Manuscript ijms-3266165-v2 "Relationship between the occurrence of depression and DROSHA (rs6877842, rs10719) and XPO5 (rs11077) single nucleotide polymorphisms in the Polish population: a case-control-study" by Mateusz Kowalczyk et al.
In this revised manuscript, the authors have now slightly improved their study. They provide evidence that using a cohort of 200 individuals, comprising 100 with depressive disorders in the study group and 100 without MDD in the control group, has sufficient power to trust their findings.
However, a weak part of this study remains about putative functions of identified polymorphisms, rs10719 in DROSHA gene and rs11077 in XPO5 gene.
The authors now propose that "miR-16 regulates the expression of the serotonin transporter, and its dysregulation has been implicated in depressive disorders [22]. Given that XPO5 is crucial for the nuclear export of pre-miRNAs, including miR-16, polymorphisms such as rs11077 could disrupt this export process, potentially leading to altered serotonin signaling and contributing to the pathogenesis of depression”.
Since serotonin transporter is expressed both in CNS and in periphery, it should be quite straightforward to assessed its expression levels in blood platelets as an example. This would add a strong value to the authors findings.
Author Response
Since serotonin transporter is expressed both in CNS and in periphery, it should be quite straightforward to assessed its expression levels in blood platelets as an example. This would add a strong value to the authors findings.
Dear Reviewer, thank you for your constructive feedback and for acknowledging the improvements in our revised manuscript. We appreciate your insights, particularly regarding the functional validation of the polymorphisms rs10719 in DROSHA and rs11077 in XPO5.
We agree that assessing the expression levels of the serotonin transporter in peripheral blood platelets could provide valuable functional insight into how these polymorphisms, particularly rs11077 in XPO5, might influence serotonin signaling pathways. Such an analysis would indeed strengthen the link between miRNA processing disruptions and serotonin transporter dysregulation, thereby enhancing the validity of our findings.
While this study focused primarily on genetic associations and the statistical power behind them, we recognize the importance of functional validation to deepen understanding of the mechanistic role of XPO5 in the nuclear export of pre-miRNAs like miR-16, which impacts serotonin transporter expression. We plan to incorporate this aspect in future research by assessing serotonin transporter levels in blood platelets to provide additional biochemical evidence supporting the genetic associations observed.
We have also incorporated future research plans in the manuscript, as suggested by the reviewer. Thank you for this valuable recommendation, which has further shaped the direction of our research.
Reviewer 3 Report
Comments and Suggestions for Authors
I appreciate the authors' respond. However, in light of the very low response rate from the tested subjects, I'm not convinced that the results are accurate.
1. One potential suggestion to the authors I have is to do the following: 1) limit the analyses to the 48 subjects (adjusting for potential demographic confounds) on who they have accurate demographic information (this will be underpowered), and then 2) attempt to correlate effect sizes from the analysis in the 48 people with effect sizes obtained from the analysis of the entire sample. If the correlation of the effect sizes is reasonable (i.e., it is positive and significant) that may increase the confidence that the results are not driven by unknown confounds. I understand it is a crude approach, and the presence of significant positive correlation is not guarantee that the results are true, however, the confidence in these results will be higher.
2. Additionally, I'd suggest that they also interrogate the latest meta-GWAS of MDD, that was just published (https://doi.org/10.1101/2024.04.29.24306535), G and see if any of the 3 SNPs they tested appear in the meta-GWAS of MDD or is in LD with the significant SNPs in the study.
Author Response
Dear Reviewer,
Thank you for your thoughtful feedback and constructive suggestions to strengthen our analysis. We appreciate your insights and will address each of your points below.
- One potential suggestion to the authors I have is to do the following: 1) limit the analyses to the 48 subjects (adjusting for potential demographic confounds) on who they have accurate demographic information (this will be underpowered), and then 2) attempt to correlate effect sizes from the analysis in the 48 people with effect sizes obtained from the analysis of the entire sample. If the correlation of the effect sizes is reasonable (i.e., it is positive and significant) that may increase the confidence that the results are not driven by unknown confounds. I understand it is a crude approach, and the presence of significant positive correlation is not guarantee that the results are true, however, the confidence in these results will be higher.
Thank you for your insightful suggestion. We initially conducted analyses limited to the subset of 48 participants with complete demographic information. However, due to the lack of statistical significance and correlation with demographic variables, we decided to abandon this approach. We also observed that the survey data provided by participants were often incomplete or of questionable accuracy, leading us to doubt their reliability.
For these reasons, we chose to increase the sample size without relying on demographic survey data. We recognize that our findings require further validation in future studies, but, unfortunately, we are currently limited in our ability to pursue additional analyses.
- Additionally, I'd suggest that they also interrogate the latest meta-GWAS of MDD, that was just published (https://doi.org/10.1101/2024.04.29.24306535), G and see if any of the 3 SNPs they tested appear in the meta-GWAS of MDD or is in LD with the significant SNPs in the study.
Thank you very much for your valuable suggestion regarding the recent meta-GWAS of major depressive disorder (MDD). We have carefully reviewed this meta-analysis, specifically investigating the three SNPs included in our study: rs6877842 and rs10719 in the DROSHA gene, and rs11077 in the XPO5 gene. To contextualize our findings within a broader genetic landscape, we referenced this meta-analysis in our publication.
Unfortunately, as the study is currently available only as a preprint, detailed tables with results are not publicly accessible. Nevertheless, even if our SNPs were not among those identified as significant for depression in this meta-analysis, our findings would still represent a unique contribution to understanding the genetic basis of depression. Our research, focuses on a specific, localized population—the Polish cohort—which may suggest the presence of genetic distinctions across ethnic groups. While our study points to associations between DROSHA and XPO5 SNPs and depression in this population, it aligns with other studies indicating these genes’ potential roles in miRNA biogenesis pathways that might be disrupted in depressive disorders. Consequently, these findings highlight the need for further comparative studies across different populations to clarify the potential role of DROSHA and XPO5 SNPs in depression.
Reviewer 4 Report
Comments and Suggestions for Authors
Please put gene names in italics
Author Response
Please put gene names in italics.
It has been corrected.
Round 3
Reviewer 3 Report
Comments and Suggestions for Authors
i have no further comments to the reviewers